# Discrete Sliding Mode Control Strategy for Start-Up and Steady-State of Boost Converter

**Tao Yang ***  **and Yong Liao**

State Key Laboratory of Power Transmission Equipment & System Security and New Technology, Chongqing University, Chongqing 400044, China

**\*** Correspondence: 20121101011@cqu.edu.cn

**Abstract:** Since the zero initial conditions of the boost converter are far from the target equilibrium point, the overshoot of the input current and the output voltage will cause energy loss during the start-up process when the converter adopts the commonly used small-signal model design control method. This paper presents a sliding mode control strategy that combines two switching surfaces. One switching surface based on the large-signal model is employed for the start-up to minimize inrush current and voltage overshoot. The stability of this strategy is verified by Lyapunov theory and simulation. Once the converter reaches the steady-state, the other switching surface with PI compensation of voltage error is employed to improve the robustness. The latter switching surface, which is adopted to regulate the voltage, can not only suppress the perturbation of input voltage and load, but also achieve a better dynamic process and a zero steady-state error. Furthermore, the discrete sliding mode controller is implemented by digital signal processor (DSP). Finally, the results of simulation, experiment and theoretical analysis are consistent.

**Keywords:** boost converter; sliding mode control; start-up; target equilibrium point; voltage regulation

## 1. Introduction

Nowadays, the research of renewable energy is increasing due to energy shortages and pressure to reduce greenhouse gas emissions [1]. Power converters play an important role in renewable energy conversion [2]. Fuel cell hybrid electric vehicles need a DC–DC boost converter to suppress the variations of the fuel cell output voltage [3,4]. The nonlinearity of the boost converter is an obstacle when we analyze its characters [5–7]. To overcome this problem, a linear model of the boost converter should be built. The state-space averaging method is a commonly used linear modeling method, that is, its linear small-signal model is established around the steady-state operating point [8]. However, in the start-up process, the initial value of output voltage and the initial value of input current are both equal to zero and there is a large deviation from the target equilibrium point. The inrush current and voltage overshoot occur during the start-up process. Moreover, there is a nonminimum phase phenomenon in the control method of a boost converter based on the small-signal model, Therefore, it is impossible to deal with the system parameter changes and large-signal transient caused by start-up or load changes [9–11].

In a practical application, the sliding mode control (SMC) has the advantages of a fast dynamic response, good stability and strong anti-disturbance ability [12–14]. A grid voltage observer based on a sliding mode is proposed in [15] and verified by experiments. The basic positive and negative sequences can be accurately estimated and separated for voltage sensorless operation in an unbalanced network. An extended Luenberger-sliding mode observer is proposed in [16], which can estimate the rotor flux and resistance along with the adaptation method to compensate for the error caused by

the mismatch between the motor parameter and the model in the controller. A novel adaptive-gain sliding mode observer is proposed in [17], which can improve the precision of sensorless control of permanent magnet linear synchronous motors. In [18], an improved sliding controller design method for the photovoltaic system is proposed, which can alleviate the disturbance caused by irradiance change and oscillation in large capacity voltage. Moreover, the controller designed by the improved method can drive photovoltaic voltage to accurately track MPPT reference value obtained from external calculation. For the quadratic boost converter, a sliding-mode controller based on a fixed-frequency pulse-width modulation (PWM) is proposed in [19]. A feed-forward control technique is proposed in [20] to improve the steady-state performance of the DC–DC asymmetric Multistage Stacked Boost Architecture converter. A sliding mode controller of the boost converter based on a robust pulse-width modulation is proposed in [21], which supplies power to constant power load in typical DC micro-grid. The methods of [19–21] do not consider the start-up situation and are only suitable for the steady-state.

The inrush current or voltage overshoot occurs in many power electronic converters during the start-up process. An input-adaptive boost converter that can start up by itself is proposed in [22], which has the advantages of strict input regulation and high efficiency. However, due to many passive devices in a boost converter, the efficiency is reduced. The inrush current analysis is given in [23,24], where the auxiliary diode and inductor of a boost converter are connected in parallel to obtain the minimum inrush current. However, although the inrush current can be reduced by an auxiliary diode branch, high $du/dt$ happens in the filter capacitor, which may damage the components and devices.

Based on the analysis given above, a sliding mode control method combining two switching surfaces is proposed for different operating conditions of the boost converter in the start-up and the steady-state in this paper. The first switching surface is based on the big-signal model for start-up processes, to reduce the inrush current and voltage overshoot. The second surface with the PI controller is adopted to regulate the output voltage and suppress the perturbation of the input voltage and load when the boost converter is in the steady-state.

The rest of this paper is arranged as follows. In Section 2, the first switching surface $S_1$ ($t$) is proposed in the start-up process of the boost converter, and its stability is analyzed. In Section 3, the other switching surface $S_2$ ($t$) is proposed to regulate the output voltage in the steady-state. After that, the proposed discrete sliding mode control strategy is implemented based on DSP in Section 4. Then the experimental results are shown and analyzed in Section 5. Finally, conclusions are summarized in Section 6.

## 2. Sliding Control Strategy for Start-Up

Figure 1a,b corresponds to the circuit states of the boost converter in the switching-on state and switching-off state of a switching tube VT, respectively. The corresponding Equations (1) and (2) of Figure 1a is as follows:

$$L\frac{di_{\mathrm{L}}}{dt} = u_{\mathrm{in}} \tag{1}$$

$$C\frac{du_{\mathrm{o}}}{dt} = \frac{u_{\mathrm{o}}}{R} \tag{2}$$

where $L$, $i_{\mathrm{L}}$, $u_{\mathrm{in}}$, $C$, $u_{\mathrm{o}}$ and $R$ represent inductor, inductor current, input voltage, capacitor, output voltage and load resistance, respectively. The corresponding Equations (3) and (4) of Figure 1b is as follows:

$$L\frac{di_{\mathrm{L}}}{dt} = u_{\mathrm{in}} - u_{\mathrm{o}}, \tag{3}$$

$$C\frac{du_{\mathrm{o}}}{dt} = i_{\mathrm{L}} - \frac{u_{\mathrm{o}}}{R}. \tag{4}$$

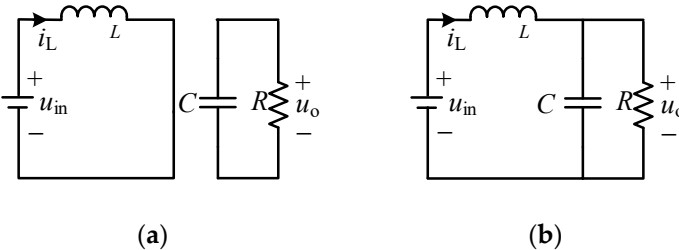

**Figure 1.** Circuit configurations of the boost converter in different switch state. (**a**) VT is switching-on. (**b**) VT is switching-off.

Variable $p$ is defined as a switching state variable, if VT is switching-on, $p = 1$; else $p = 0$, the equations of boost converter can be expressed as:

$$L\frac{di_L}{dt} = u_{in} - (1-p)u_o, \tag{5}$$

$$C\frac{du_o}{dt} = (1-p)i_L - \frac{u_o}{R}. \tag{6}$$

## 2.1. Sliding Function for Start-Up

The average input power of the battery in one switching period is equal to the output power of load when the boost converter operates at a steady state. Ignoring the internal resistance and voltage drop of tube and diode, the relationship between $i_L$ and $u_o$ at steady-state is satisfied as follows:

$$i_L = \frac{u_o^2}{u_{in}R}u_o > u_{in}. \tag{7}$$

Due to $u_o > u_{in}$, the trajectory of Equation (7) is a part of a parabola. Define $S_1\,(t)$ as the switching surface for the start-up process, as shown below:

$$S_1(t) = I_Lu_o(t) - U_oi_L(t). \tag{8}$$

$(I_L, U_o)$ is the target equilibrium point satisfying Equation (7) at steady-state. $I_L$ and $U_o$ represent the inductor current and output voltage at the target equilibrium point respectively. The sliding mode control law is shown as:

$$\begin{aligned} p &= 1 \text{ if } S_1(t) > 0 \\ p &= 0 \text{ if } S_1(t) < 0 \end{aligned} \tag{9}$$

Figure 2 shows the sliding mode control model for start-up process, $i_L$ and $u_o$ are detected in the sampling period and substitute into Equation (8), so $S_1$ can be obtained. Finally, the switching signal of VT is generated by Equation (9).

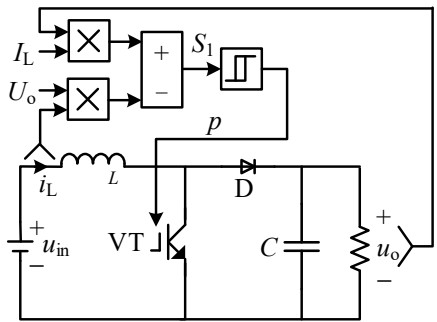

**Figure 2.** The sliding-mode control model for the start-up process.

## 2.2. Stability Analysis of the Sliding Function

Define the Lyapunov function as follows:

$$P = \frac{1}{2}S_1^2, \tag{10}$$

where the derivative of $P$ is:

$$\dot{P} = S_1 \cdot \dot{S}_1. \tag{11}$$

According to Equations (5), (6) and (8), the expression of $\dot{S}_1$ can be obtained:

$$\dot{S}_1 = \frac{I_L}{C}\left(i_L - pi_L - \frac{u_o}{R}\right) - \frac{U_o}{L}(u_{in} - u_o + pu_o). \tag{12}$$

According to Equation (9), if $S_1 > 0$, $\dot{S}_1$ can be rewritten as:

$$\dot{S}_1 = -\frac{I_L u_o}{RC} - \frac{u_{in} U_o}{L}. \tag{13}$$

Due to $u_o > u_{in} > 0$, so $\dot{S}_1 < 0$.
If $S_1 < 0$, $\dot{S}_1$ is:

$$\dot{S}_1 = \frac{I_L}{C}\left(i_L - \frac{u_o}{R}\right) - \frac{U_o}{L}(u_{in} - u_o). \tag{14}$$

According to the characteristics of voltage boost and current buck, $i_L > u_o/R$, so $\dot{S}_1 > 0$.

Based on the above analysis, regardless of the value of $S_1$, $P\dot{P} < 0$ can be satisfied. Therefore, the proposed sliding mode control system is stable in this paper, $i_L$ and $u_o$ can move to the trajectory of $S_1$ from any initial state, then reach the equilibrium point $(I_L, U_o)$ along the switching surface.

## 2.3. Simulation of the Sliding Mode Control Based on $S_1$ in the Start-Up Process

The parameters of the boost converter adopted in this paper are shown in Table 1, where the values of inductance and capacitance are satisfied as follows:

$$L > \frac{u_{in}D_{max}}{\Delta i_{Lmax}f_s}, \tag{15}$$

$$C > \frac{D_{max}}{Rf_s\eta_{max}}, \tag{16}$$

where $D_{max}$ is the max duty, $f_s$ is switching frequency, $\Delta i_{Lmax}$ is the max current ripple, and $\eta_{max}$ is the max voltage ripple ratio. After setting $D_{max} = 0.8$, $\Delta i_{Lmax} = 0.5$ A, $\eta_{max} = 1\%$, $f_s = 10$ kHz, Equations (15) and (16), can get the range of $L$ and $C$ is $L > 1.9$ mH and, $C > 192$ μF, respectively. In this paper, $L$ take the value of 2 mH and $C$ is set to 265 μF.

**Table 1.** Main parameters for the boost converter in the simulation and experiment.

| Parameters | Values |
|---|---|
| Input Voltage $u_{in}$ | 12 V |
| Input Inductance $L$ | 2 mH |
| Filter Capacitance $C$ | 265 μF |
| Resistance Load $R$ | 50 Ω |

The target equilibrium point is set as $I_L = 1.02$ A, $U_o = 24$ V. Figure 3a shows the trajectory from original point (0 A, 0 V) to the target equilibrium point $(I_L, U_o)$ by Matlab. The motion process of $i_L$

and $u_o$ is divided into two steps: Firstly, they deviate from $T_{S1}$ and increase simultaneously until $u_o > u_{in}$; then they start to approach $T_{S1}$. After that, when they reach $T_{S1}$, they move along $T_{S1}$ to the target equilibrium point ($I_L$, $U_o$). Even if the initial point is not at the original point, $i_L$ and $u_o$ can still adjust to the target equilibrium point by $S_1$. The trajectory from the initial point (2.5 A, 30 V) to the target equilibrium point is shown in Figure 3b.

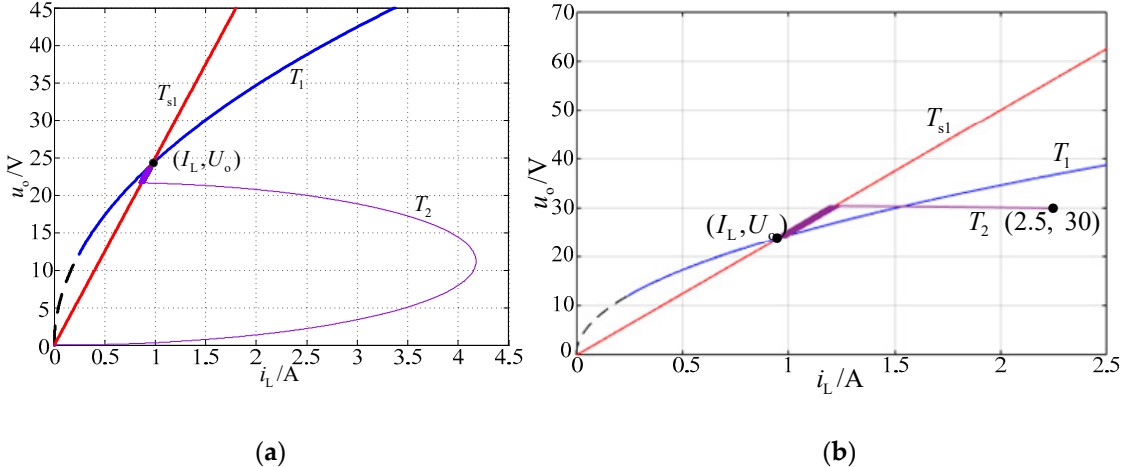

**Figure 3.** Start-up trajectory in the $i_L$-$u_o$ plane. $T_{S1}$ is the sliding curve of $S_1$, $T_1$ is the trajectory of (7), $T_2$ is the tracking of the motion of $i_L$ and $u_o$, respectively. (**a**) The initial point is (0 A, 0 V). (**b**) The initial point is (2.5 A, 30 V).

Figure 4a shows the simulation results of the control strategy based on the small-signal model in the start-up process. For comparison, Figure 4b shows the simulation results of the sliding mode control strategy proposed in this paper. In Figure 4a, the peak of $i_L$ and $u_o$ is corresponding to 5.86 A and 28.75 V, respectively. However, there is no voltage overshoot in Figure 4b, and the peak of $i_L$ is only 4.12 A, which is lower than $i_L$ shown in Figure 4a. Moreover, current discontinuation occurs in Figure 4a, therefore the time from the zero initial state to a steady state is much longer than Figure 4b. The results show that the sliding mode control of $S_1$ is correct and feasible.

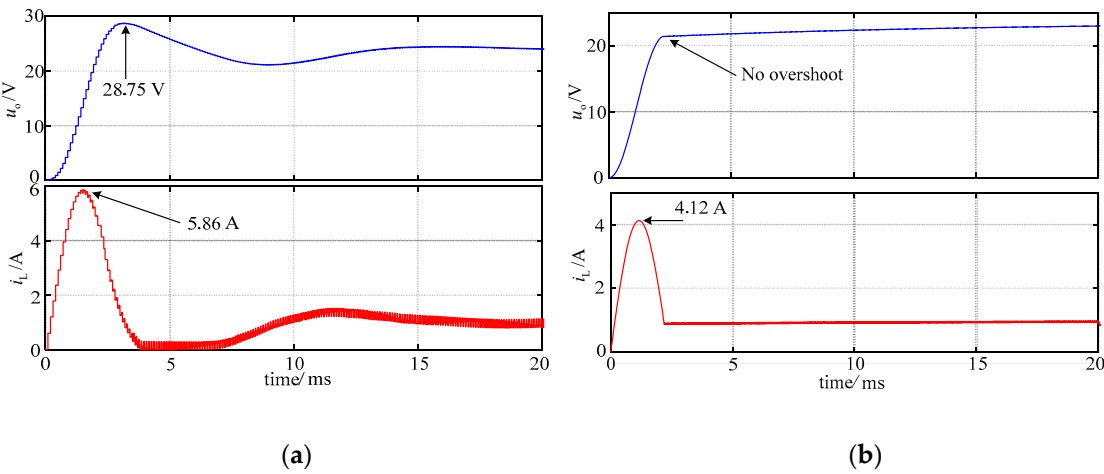

**Figure 4.** The simulation result of $i_L$ and $u_o$ in the start-up process. (**a**) Control based on the small-signal model. (**b**) Sliding control based on the big-signal model.

## 3. Voltage Control Strategy for Steady-State

Since the control strategy based on $S_1$ has a long time along the sliding line and poor robustness, it is only suitable for the start-up process. During steady-state, an error will occur when $u_{in}$ or $R$

changes, therefore, another control strategy should be adopted to regulate the output voltage at this stage. Define switching surface $S_2$ ($t$) as follows:

$$S_2(t) = I_L + \Delta i_L(t) - i_L(t) \tag{17}$$

where $\Delta i_L$ is current error compensation. When $u_{in}$ or $R$ is perturbed, if $U_o$ is constant, the target equilibrium point changes from ($I_L$, $U_o$) to ($I_L + \Delta i_L$, $U_o$). $\Delta i_L$ can be expressed as

$$\Delta i_L(t) = k_p(U_o - u_o(t)) + k_i \int_{t_1}^{\infty} U_o - u_o(t)dt \tag{18}$$

where $k_p$ is the proportional gain and $k_i$ is the integral gain of the voltage PI controller. $t_1$ is the moment that the switching surface is switched from $S_1$ to $S_2$. Substituting Equation (18) into Equation (17), $S_2$ ($t$) can be expressed as:

$$S_2(t) = I_L - i_L(t) + k_p(U_o - u_o(t)) + k_i \int_{t_1}^{\infty} U_o - u_o(t)dt. \tag{19}$$

The sliding control law of $S_2$ is shown as:

$$\begin{aligned} p &= 1 \text{ if } S_2(t) > 0 \\ p &= 0 \text{ if } S_2(t) < 0 \end{aligned} \tag{20}$$

Figure 5 shows the discrete sliding mode control model for steady-state. $\Delta i_L$ is obtained by discrete PI operation of voltage error $u_e$, then the switching signal of VT is generated by Equation (20). $z^{-1}$ is the delay unit, which is used to implement the discrete integral operation.

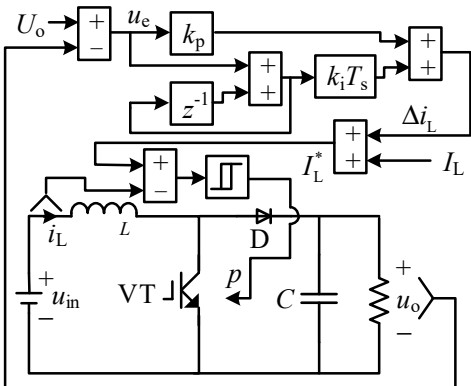

**Figure 5.** The discrete sliding mode control model for steady-state.

### 3.1. Analysis of Transient Process under Input Voltage Perturbations

Figure 6 shows the transient trajectory of $i_L$ and $u_o$ under different input voltage perturbations. The increase or decrease of $u_{in}$ will lead to the deviation of the equilibrium point locus (EPL). Therefore, the steady-state equilibrium operating point will change from O ($I_L$, $U_o$) to Q ($I_{LQ}$, $U_o$) or P ($I_{LP}$, $U_o$). $I_{LP}$ and $I_{LQ}$ represent the inductor current at P and Q points respectively. When $u_{in}$ steps from $U_1$ to $U_3$ ($U_1 > U_3$), ($i_L$, $u_o$) immediately move from initial equilibrium operating point O ($I_L$, $U_o$) to Q$_1$. The new reference value of $i_L$ can be obtained by PI operation of voltage error and generates the VT switching signal according to the sliding control law of $S_2$, makes $i_L$ equal to $I_{LQ}$ and $u_o$ recovers to $U_o$ correspondingly due to power conservation. A symmetrical situation will occur when $u_{in}$ steps from

$U_1$ to $U_2$. As shown in the same figure, the correction effects make $i_L$ equal to $I_{LP}$. Along the trajectory O→$P_1$→P, ($i_L$, $u_o$) reach to the new equilibrium point P ($I_{LP}$, $U_o$).

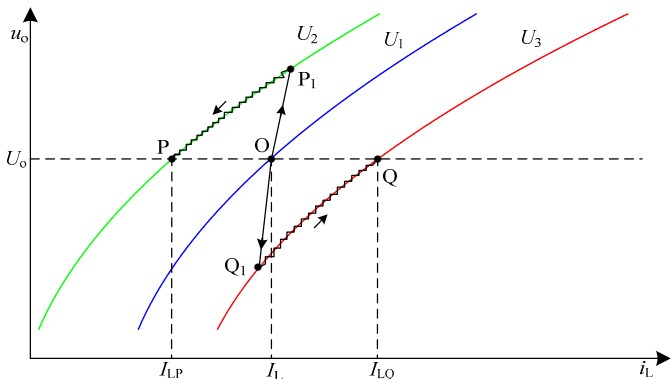

**Figure 6.** The transient trajectory in the $i_L$-$u_o$ plane for input voltage perturbations.

### 3.2. Analysis of Transient Process under Load Perturbations

Figure 7 shows the transient trajectory of $i_L$ and $u_o$ under different load perturbations. When $R$ decreases from $R_0$ to $R_1$, the transient output power is larger than the input power, which results in $u_o$ decreases and $i_L$ increases. Therefore, ($i_L$, $u_o$) immediately move from the initial equilibrium operating point O ($I_L$, $U_o$) to $M_1$, then reach the new equilibrium operating point M ($I_{LM}$, $U_o$) under the control of the sliding law of $S_2$. A symmetrical situation will occur when load resistance $R$ increases from $R_0$ to $R_2$. As shown in the same figure, the correction effects make $i_L$ equal to $I_{LN}$. Along the trajectory O→$N_1$→N, ($i_L$, $u_o$) reach to the new equilibrium point N ($I_{LN}$, $U_o$).

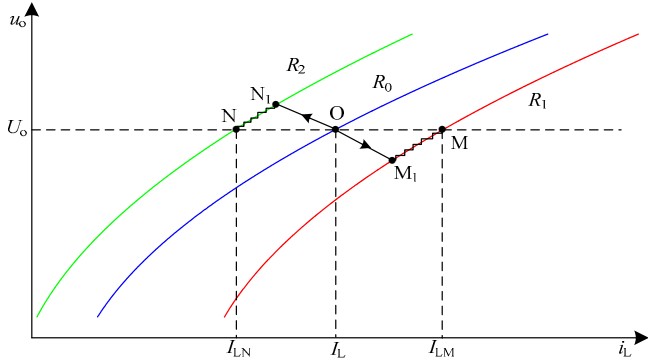

**Figure 7.** The transient trajectory in the $i_L$-$u_o$ plane for load perturbations.

### 3.3. Simulation of the Sliding Control Based on $S_2$ in Steady-State

Figure 8 shows the simulation result of the transition between switching sliding function $S_1$ and $S_2$ when the $u_o$ increases to $U_o$ (24 V), there is only a small dynamic voltage deviation from $U_o$ due to the PI regulator of $u_o$. As shown in Figure 8, the duration of the start-up is 13 ms. The ripple of $u_o$ is less than 0.05 V.

Figure 9 shows the simulation response of the step perturbation superimposed on the input voltage. The nominal value of the input voltage is 12 V in Figure 9a. $u_{in}$ is reduced from 12 V to 9 V at 0.15 s, which leads to a deviation of $u_o$. However, it only takes 22 ms for $u_o$ to recover to nominal value due to the sliding mode control based on $S_2$. Moreover, the maximum reduction value is only 1.28 V. Figure 9b also shows good dynamic and steady-state performance when $u_{in}$ increases from 12 V to 15 V. The transient process is consistent with the ($i_L$, $u_o$) trajectory shown in Figure 6.

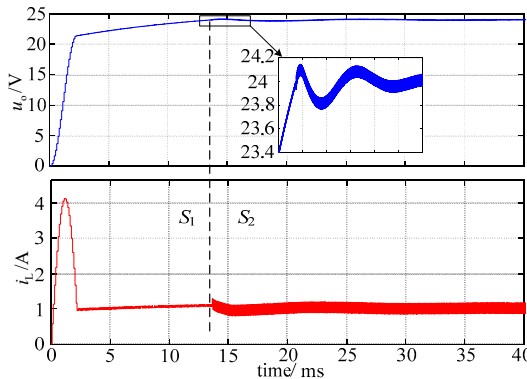

**Figure 8.** Simulation results of the transition between the two sliding functions.

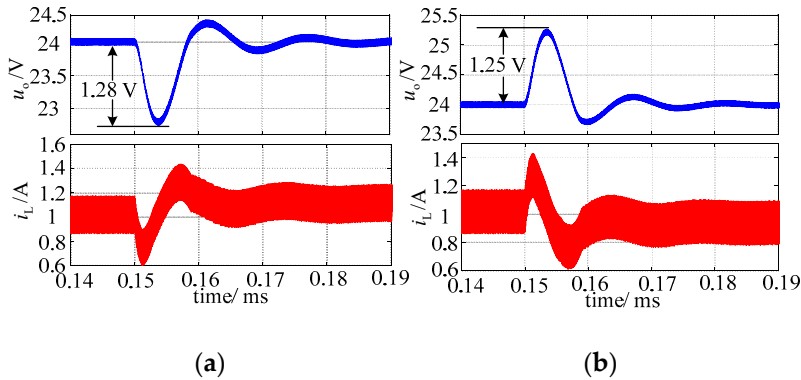

(**a**)          (**b**)

**Figure 9.** Simulation response to input voltage perturbations. (**a**) Reduction of $u_{in}$. (**b**) Increase of $u_{in}$.

Figure 10 shows the simulation responses of $u_o$ and $i_L$ under the perturbation of load from 40 $\Omega$ to 50 $\Omega$ and from 50 $\Omega$ to 40 $\Omega$ respectively. After a slight transient oscillation, $u_o$ returned to the nominal value of 24 V. The maximum reduction value is 0.7 V and dynamic adjustment time is 15 ms. The transient process is consistent with the track of ($i_L$, $u_o$) shown in Figure 7.

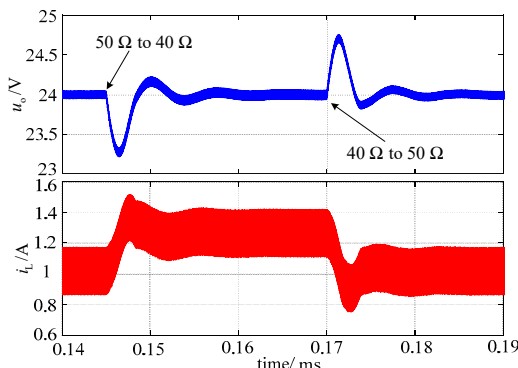

**Figure 10.** Simulation response to load perturbations.

## 4. Electronic Implementation Based on DSP

Figure 11 is a photograph of the boost converter experimental platform. In this platform for experimental verification, an oscilloscope is used to distinguish the voltage. The input voltage is supplied by a programmable power source. The type of DSP is TMS320F28335, which supports floating-point operation. The type of IGBT is PM400HSA120 and the type of diode is RM300HA-24F, which have a sample frequency up to 40 kHz.

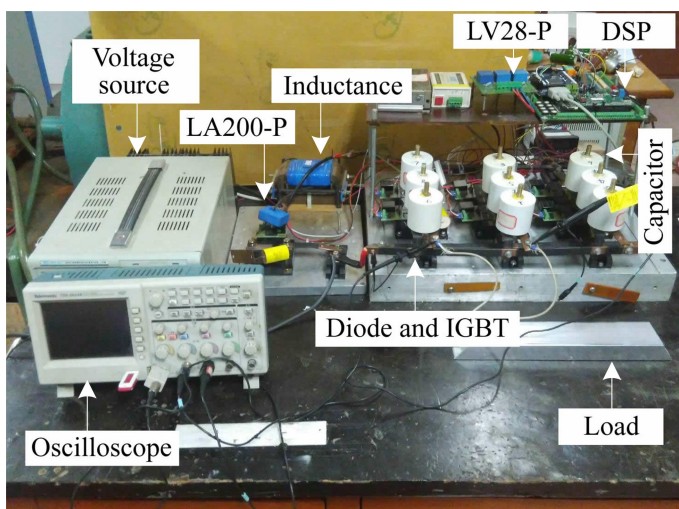

**Figure 11.** Picture of the experimental setup for the boost converter.

### 4.1. Detection and Conversion of Input Current and Output Voltage

Figure 12 shows the detection and filter circuit of $i_L$ and $u_o$, current sensor (LA200-P) and voltage sensor (LV28-P) are adopted to detect $i_L$ and $u_o$, respectively. The output current $i_{IM}$ of LA200-P is proportional to $i_L$.

$$i_{IM} = k_1 i_L \tag{21}$$

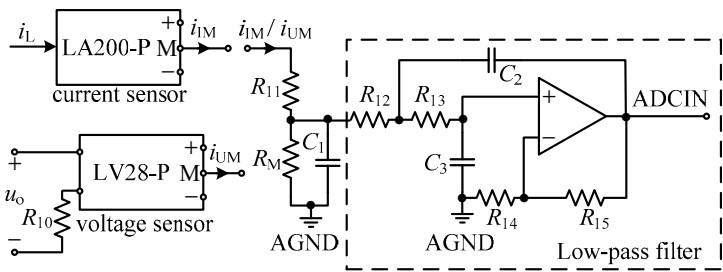

**Figure 12.** Detection and filter circuit of $i_L$ and $u_o$.

$k_1$, which is the conversion ratio of LA200-P, equal to 0.0005. For voltage detection, an input resistance $R_{10}$ is used to convert the voltage to the primary current, so the relationship between the output current $i_{UM}$ of LV28-P and $u_o$ is

$$i_{UM} = k_2 u_o / R_{10}. \tag{22}$$

$k_2$ is the conversion ratio of LV28-P and equal to 2.5, then $i_{UM}$ and $i_{IM}$ generate voltage across the sample resistance $R_M$. Before enforcing A/D conversion by DSP, the voltage of $R_M$ should be filtered by a low pass filter circuit with a bandwidth of 200 Hz.

Since the size of the result register in TMS28335 is 12-bits and the maximum conversion analog voltage is 3 V, the digital value ($X_{IM}$) of the result register used to obtain $i_L$ can be expressed as:

$$X_{IM} = (2^{12} - 1) i_{IM} R_M / 3. \tag{23}$$

According to Equations (21) and (23), DSP can calculate $i_L$ from $X_{IM}$ by:

$$i_L = \frac{3 X_{IM}}{(2^{12} - 1) k_1 R_M}. \tag{24}$$

The calculation method of $u_o$ from the digital value ($X_{UM}$) of result register is given directly:

$$u_o = \frac{3X_{IM}R_{10}}{(2^{12} - 1)k_2R_M}.$$

(25)

*4.2. Software Implementation for Sliding Mode Control*

The discrete program of sliding mode control based on $S_1$ and $S_2$ is enforced in the interrupt program caused by A/D conversion. A Boolean type variable flag is created to determine whether $u_o$ is in the steady-state. If flag = 0, $u_o$ is still in the start-up process, so the switching signal generated by the discrete program of sliding mode control is based on $S_1$. Otherwise, $S_2$ is adopted to replace $S_1$. The flow chart is shown in Figure 13.

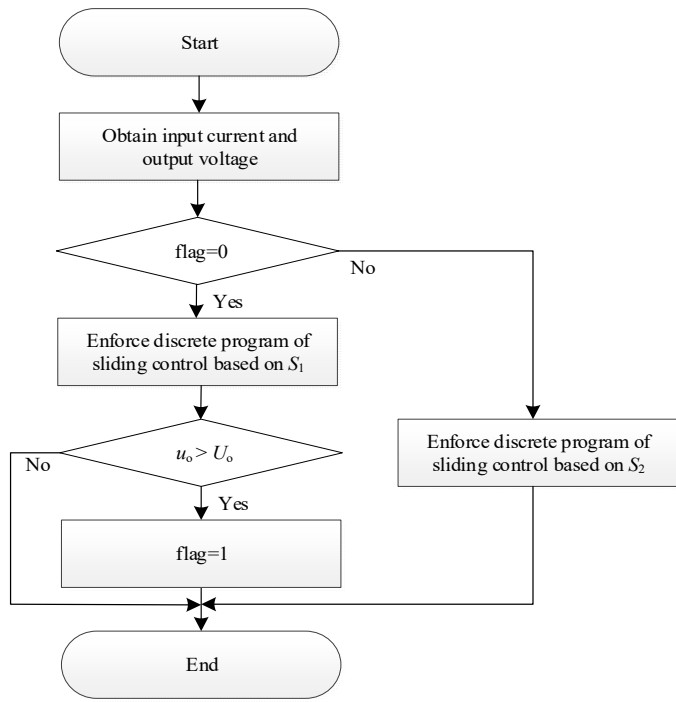

**Figure 13.** Interrupt program flow chart caused by A/D conversion.

## 5. Experimental Results

Figure 14 shows the experimental waveform of $u_o$ from the zero initial-state to the steady-state. Compared to the results shown in Figure 4a, there is no overshoot of $u_o$ in the start-up process under the effect of sliding mode control based on $S_1$. When $u_o$ first increases to 24 V, sliding mode control based on $S_2$ is adopted, the transition between switching surface $S_1$ and $S_2$ is in an agreement with the simulation results shown in Figure 8. As shown in this figure, good performance of the boost converter in the start-up process can be obtained by the switching surface $S_1$. The duration of the start-up is 20 ms and the voltage overshoot is only 4.1%.

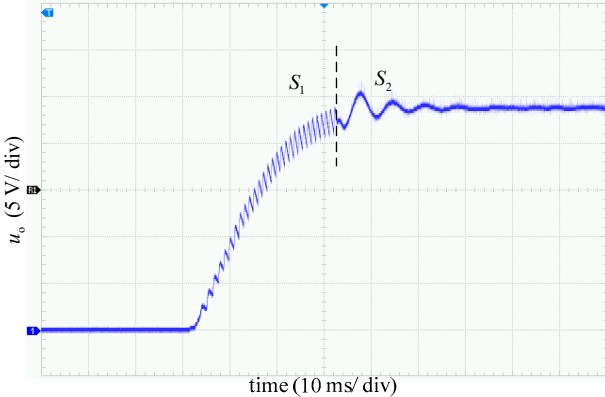

**Figure 14.** Experimental results from the zero steady-state to the steady-state.

Figure 15 illustrates the experimental response of the input voltage step decreasing from 12 V to 9 V. In contrast, Figure 16 illustrates the response of the input voltage, increasing from 12 V to 15 V. The experimental results are in good agreement with the simulation results shown in Figure 9. After ignoring the transient state, the action of the switching surface $S_2$ can realize a zero steady-state error of $u_o$. The maximum deviation value is 1.3 V and the dynamic adjustment time is less than 20 ms.

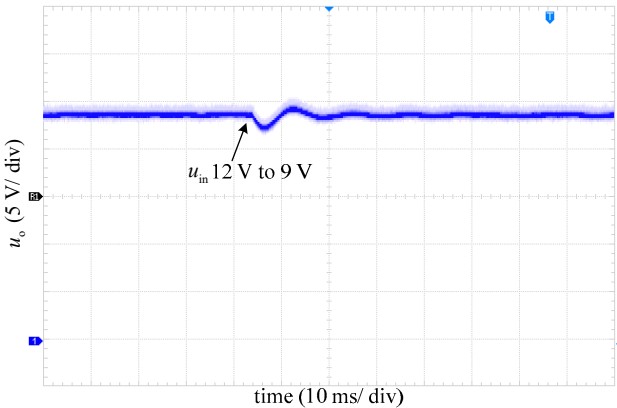

**Figure 15.** Experimental response of input voltage step increasing.

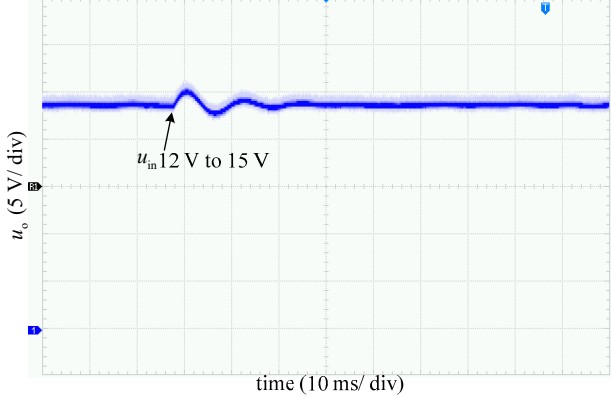

**Figure 16.** Experimental response of input voltage step decreasing.

Figure 17 shows the experimental response of the perturbation of load from 40 Ω to 50 Ω and from 50 Ω to 40 Ω respectively. A resistance of 200 Ω series connected with an auxiliary tube is parallel in the nominal load. The perturbation of load is achieved by controlling the auxiliary tube by the switching-on or switching-off state, the switching signal of the auxiliary tube is shown in Figure 17.

Under the action of the switching surface $S_2$, A zero steady-state error of $u_o$ is also achieved after ignoring the transient state. The maximum deviation value is 2.5 V and the dynamic adjustment time is less than 22 ms. The robustness is proved by simulation and experimental results.

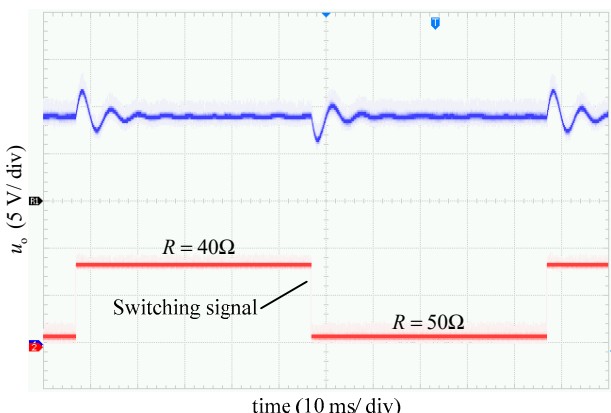

**Figure 17.** Experimental response to load perturbations.

## 6. Conclusions

For the different conditions of the start-up process and steady-state process of the boost converter, two switching surfaces are proposed in this paper respectively. In the start-up process of the boost converter, switching surface $S_1$, which based on the large-signal model to minimize inrush current, voltage overshoot and adjustment time, is adopted. When the converter operates at a steady-state, the other switching surface $S_2$ including PI compensator is employed to regulate the output voltage, which can suppress the perturbation of input voltage and load fluctuation on the output voltage. Furthermore, the effectiveness of the control strategy proposed in this paper is verified by simulation and experiment. The research results can provide a reference for the engineering design of the sliding mode boost controller, so that the minimum value of the inrush current is compatible with the high voltage regulation value. In electric vehicle applications, the sliding mode control strategy proposed in this paper can be used in a boost converter between the battery and the inverter to suppress battery voltage fluctuations and improve system efficiency.

**Author Contributions:** Investigation, T.Y.; Software, T.Y.; Writing—original draft preparation, T.Y.; Writing—review and editing, T.Y. and Y.L.

**Funding:** This research received no external funding.

**Conflicts of Interest:** The authors declare no conflict of interest.

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
