# Peer review of "Discrete Sliding Mode Control Strategy for Start-Up and Steady-State of Boost Converter"

_energies, doi:10.3390/en12152990_

Round 1

Reviewer 1 Report

In this paper, two different sliding mode control surfaces are proposed for start-up stage and steady-state stage of boost converter respectively. 

The paper is well written. However, I have the following comments.

1) Introduction should be more summarized. 

2) The discussion of the results is minimal and should be significantly expanded.

3) The paper needs a comprehensive review for grammar and language correctness.

4) Add some of the most important quantitative results to the Abstract.      

5) Cite and review the following related references:

Design and implementation of smart uninterruptable power supply using battery storage and photovoltaic arrays

Dynamic and Hybrid Phase Shift Controller for Dual Active Bridge Converter

Author Response

Response to Reviewer 1 Comments

In this paper, two different sliding mode control surfaces are proposed for start-up stage and steady-state stage of boost converter respectively.

Point 1: Introduction should be more summarized.

Response 1: The introduction has been revised. References that are not very relevant to this article are removed. Moreover, some the expressions in the introduction are more accurate and summarized.

Point 2: The discussion of the results is minimal and should be significantly expanded.

Response 2: More detailed descriptions of simulation and experimental results have been added to expand the discussion.

Point 3: The paper needs a comprehensive review for grammar and language correctness.

Response 3: Grammar and language correctness of this paper have been revised comprehensively. I apologize for my poor English and any inconveniences this may have troubled you.

Point 4: Add some of the most important quantitative results to the Abstract.

Response 4: Some descriptions of the most important quantitative results has been added to the Abstract of the article.

Point 5: Cite and review the following related references:

- Design and implementation of smart uninterruptable power supply using battery storage and photovoltaic arrays

- Dynamic and Hybrid Phase Shift Controller for Dual Active Bridge Converter

Response 5: The dynamic model and hybrid phase-shift control of a dual-active-bridge converter” has been reviewed and cited as reference [7]. Unfortunately, I have not found the other one “Design and implementation of smart uninterruptible power supply using battery storage and photovoltaic arrays”.

Thank you for your comments concerning my manuscript.

Reviewer 2 Report

The paper presents an interesting topic regarding control strategy of boost converter. The quality presentation it's high. The mathematical model it is follows by simulation part and experimental results.

Equations 15 and 17 must be completed.

Also the comparison between simulation and experimental results must be detailed.

Author Response

Response to Reviewer 2 Comments

The paper presents an interesting topic regarding control strategy of boost converter. The quality presentation it's high. The mathematical model it is follows by simulation part and experimental results.

Point 1: Equations 15 and 17 must be completed.

Response 1: Equations 15 and 17 have been completed. The calculation order of the Equations has been adjusted. And the corresponding description has been added. (The numbers of original Equations 15 and 17 has been adjusted to Equations 19 and 18 because some new Equations has been added)

Point 2: Also the comparison between simulation and experimental results must be detailed.

Response 2: This part has been revised by adding descriptions and comparisons of key indicators (such as start-up duration time, voltage ripple and maximum deviation value, dynamic adjustment time, etc.) of simulation and experimental results.

Thank you for your comments concerning my manuscript.

Reviewer 3 Report

This manuscript presents two sliding mode control strategies for different operating conditions of the boost converter in the start-up and the steady state. One strategy based on the large-signal model for start-up process to reduce the inrush current and voltage overshoot. Another strategy with PI controller is adapted to regulate the output voltage and suppress the perturbation of the input voltage and load when the boost converter is in the steady state.

The paper contains some interesting information regarding DC-DC boost converter control, however, the scientific novelties are very doubtful for me (converter topology, control approaches are well-known) and practical importance is limited (there is no connection for real application and not enough details on experimental implementation).

There are also some drawbacks listed below:

1. This statement is confusing: “The inrush current or voltage overshoot occurs in many power electronic converters during start-up process, such as voltage source inverter [1], current source inverter [18], buck converter [19], and boost converter [20-22].”. Firstly, the comparison of DC-AC with DC-DC converters is incorrect in my opinion. Secondly, [18] actually devoted to DC-DC and [21] to DC-AC applications.

2. “Figure 2. The sliding-mode control model for start-up process” is confusing. Please explain how the control could be implemented without any scaling blocks. Please explain also in more details the structure depicted in Figure 5.

3. The parameters of boost converter shown in Table 1 require detail explanations. Why these voltage level has been chosen? What is the application aimed for this boost converter? How the inductance and capacitance values were chosen? Why your case study limited to approximately 12 W power?

4. Regarding experimental verification, please explain why there are 9 capacitors in converter shown in photograph? Please justify, why the overall size of the prototype for just 12 W power is much larger than oscilloscope and voltage source?

5. Diode and IGBT have been used in the prototype. However, MOSFET power switch has been shown in all previous figures. Please provide for information on semiconductor devices used in prototype. What are the switching and sampling frequencies? And how they were chosen?

6. The conclusion claims that “In this paper, two different sliding mode control surfaces are proposed for start-up stage and steady-state stage of boost converter respectively”. However, I did not find exactly the control surfaces in the paper. Please provide and explain in details.

7. I also suggest authors to check typos and improve English grammar of the manuscript, including but not limited to: “renewable energy has been more and more applications and research”, “boost converter is a nonlinear system, a small signal model is derived”, “small signal model of boost converter, the classic control technology”, “DC-DC boost converter”, “dc/dc boost converter”, “when boost converter in the steady state”, “Moreover current discontinuous occurs in figure 4(a),”, “input volatage perturbations”, “Simulation for the the sliding” etc.

Round 2

Reviewer 3 Report

The paper was significantly improved. The mistakes have been corrected. The authors have provided detailed explanations of the main concerns. The necessary details regarding experimental results have been added to paper as well as some valuable quantitative details added to the discussion part. In general, in my opinion, the manuscript could be accepted for publication.